# Methylthiosulfonate-Based Cysteine Modifiers as Alternative Inhibitors of Mercurial-Sensitive Aquaporins

**DOI:** 10.3390/cells12131742

**Published:** 2023-06-28

**Authors:** Katrin Jeuken, Emmi Jaeger, Emily Matthews, Eric Beitz

**Affiliations:** Department of Pharmaceutical and Medicinal Chemistry, Pharmaceutical Institute, Christian-Albrechts-University of Kiel, Gutenbergstr. 76, 24118 Kiel, Germany; kjeuken@pharmazie.uni-kiel.de (K.J.); stu223839@mail.uni-kiel.de (E.J.); stu217315@mail.uni-kiel.de (E.M.)

**Keywords:** aquaporin, water, glycerol, permeability, inhibition, mercurial, methylthiosulfonate, cysteine modification

## Abstract

(1) Background: Several members of the ubiquitous aquaporin family, AQP, of water and neutral solute channels carry a cysteine residue in the selectivity filter region. Traditionally, toxic mercury-containing compounds are used to bind to the cysteine as covalent AQP inhibitors for physiological studies or analysis of structure–function relationships. (2) Methods: We tested thiol-reactive methylthiosulfonate reagents, MTS, as alternative Cys modifiers for AQP inhibition. Three MTS reagents transferring S-alkyl moieties of increasing size, i.e., S-methyl, S-*n*-propyl, and S-benzyl, were used with yeast-expressed water-selective AQP1 and the aquaglyceroporin AQP9. Respective Cys-to-Ala variants and mouse erythrocytes that naturally express AQP1 and AQP9 served as controls. (3) Results: Both wildtype AQP isoforms were inhibited by the Cys modifiers in a size-dependent manner, whereas the Cys-to-Ala-variants exhibited resistance. Sub-millimolar concentrations and incubation times in the minute range were sufficient. The modifications were reversible by treatment with the thiol reagents acetylcysteine, ACC, and dithiothreitol, DTT. (4) Conclusions: MTS reagents represent a valid alternative of low toxicity for the inhibition of mercurial-sensitive AQPs.

## 1. Introduction

Inhibition by mercurials is a hallmark of many aquaporin water and solute channels, AQP [1]. Even before the discovery of the AQPs, the water permeability of erythrocytes was shown to be strongly inhibited by the organomercurial *p*-chloromercuribenzoate, pCMBS [2]. The combination of a higher water diffusion rate than that of plain lipid membranes and a low activation energy of <5 kcal mol^−1^ hinted at the presence of proteinaceous channel entities in the erythrocyte plasma membrane. The later identified prototypical human AQP1 water channel exhibited these properties when expressed in *Xenopus laevis* oocytes [3].

AQPs are ancient and present in all kingdoms of life [4]. They share a common protein fold consisting of six transmembrane spans and two so-called half-helices that meet in the center of the lipid bilayer to form a pseudo-seventh transmembrane domain [5]. AQPs form homotetramers with an individual pore in each protomer. Depending on the pore diameter [6], selectivity is high for water, e.g., orthodox AQP1 (≈2.4 Å), or diffusion of larger uncharged solutes is facilitated, e.g., aquaglyceroporin AQP9 (>3.5 Å). Some peculiar AQPs exist that allow ions or gases to pass the tetramer [7]. Two structural features within the AQP channel define filter regions, i.e., the Asn-Pro-Ala caps at the positive ends of the half-helices forming the central NPA region, and the aromatic/arginine selectivity filter, ar/R, closer to the extracellular entry site. Both are involved in the exclusion of charged substrates [8,9]; the ar/R filter forms the narrowest constriction in the channel and additionally selects by size [10]. The ar/R regions of many AQPs carry a cysteine residue that is accessible for mercurial binding and channel blocking via its thiol moiety [1]; see Figure 1 for mammalian AQP1 (Cys189) [11] and AQP9 (Cys213) [12].

Since their discovery, the potential of AQPs as drug targets is considered high because AQP functionality is connected with a plentitude of pathophysiological situations [14,15]. These include edema, general ion/water homeostasis, cell motility of tumor cells, lipid metabolism in fatty tissue (adipositas) and liver (steatosis), or more recently immune modulation to name a few [14]. However, the development of drug-like inhibitors particularly for water-selective AQPs remains challenging due to their narrow pore diameters [15]. There was some success, though, in the development of high-affinity inhibitors of the wider aquaglyceroporins [16,17]. In the fourth decade of AQP research, tools for the modulation of human and other AQPs remain important to evaluate their role in certain physiological or pathophysiological settings. Due to the lack of potent inhibitors for most AQPs, mercurials are widely used, see for instance [18,19,20,21,22,23,24,25,26], despite their toxicity and environmentally unfriendly properties [27].

In this study, we show that methylthiosulfonate-based cysteine modifiers provide a safe and effective alternative to mercurials for AQP inhibition. We treated yeast-expressed mammalian AQP1 and AQP9 with Cys modifiers of increasing size, i.e., S-methyl-methylthiosulfonate (methyl-MTS, MMTS), S-*n*-propyl-MTS (PMTS), and S-benzyl-MTS (BMTS). BMTS and the reference mercurial pCMBS fully inhibited AQP1 water permeability. MMTS and PMTS elicited intermediate levels of inhibition, which can be useful if modulation of AQP functionality is desired. The Cys-modifications were reversible by breaking the formed disulfide bonds by the thiol reagents acetylcysteine (ACC) and dithiothreitol (DTT). The MTS reagents were tolerated by mammalian cells and effectively blocked AQP1 and AQP9, facilitating water and glycerol permeability of mouse erythrocytes. By using a Cys213Ala variant of AQP9, we found that the earlier proposed Cys213 is not the only binding site responsible for channel blockage, but other cysteine residues of the channel path can be accessed by the modifiers as well.

## 2. Materials and Methods

Plasmids, cloning, and mutations. The previously described rat AQP1 (NCBI Gene ID 25240) and an AQP1 Cys189Ala variant [28] were expressed in yeast from the plasmid pRS426MET25; human AQP9 (NCBI Gene ID 366) [29] was expressed from the pDR196 vector. Site-directed mutagenesis was carried out using the QuikChange protocol (Agilent) to generate an AQP9 Cys213Ala variant using the forward primer 5′-GGA CTG AAC AGT GGC GCT GCC ATG AAC CCA CT-3′ (mutation site underlined) and the reverse primer 5′-GCC ACT GTT CAG TCC CAG GGA GGA AGC AAT-3′. All constructs were verified by DNA sequencing.

Yeast culture and transformation. W303-1A jen1Δ ady2Δ (MATa, can1-100, ade2-loc, his3-11-15, leu2-3,-112, trp1-1-1, ura3-1, jen1: kanMX4, ady2::hphMX4) *Saccharomyces cerevisiae* yeast was used [30]. The cells were transformed using the lithium acetate/single-stranded carrier DNA method [31]. Cells were grown at 29 °C in a selective SD medium supplemented with adenine, histidine, leucine, and tryptophan, but lacking uracil.

Permeability assays by stopped-flow light scattering. Yeast cells in liquid culture were harvested at an OD_600_ of 1 (2000× *g*, 4 °C, 10 min) and resuspended in 10 mM MOPS buffer, pH 7.2, supplemented with 50 mM NaCl, 5 mM CaCl_2_, and 1.2 M sucrose. Walled yeast cells without protoplastation were used for the assays. Mouse erythrocytes (experiments were approved by the Amt für Gesundheit und Verbraucherschutz and the Ministerium für Landwirtschaft, Umwelt und ländliche Räume des Landes Schleswig-Holstein) were washed and harvested (2000× *g*, 4 °C, 10 min) in 10 mM HEPES buffer, pH 7.4, plus 154 mM NaCl, 5 mM KCl, 5 mM glucose, and 0.1 % BSA. The cell suspensions were adjusted to an OD_600_ of 2 for the assays. For inhibition, solutions in DMSO of MMTS (Sigma-Aldrich, Taufkirchen, Germany), PMTS (Interchim, Montluçon, France), BMTS (Biozol, Eching, Germany) or pCMBS (kindly provided by M. Bleich) were added to the cells (1% DMSO final) 10 min prior to the assay to achieve final concentrations of 30–1000 µM of the modifying agent as indicated. For reversal of the Cys-modification by ACC (Fagron, Glinde, Germany) or DTT (Roth, Karlsruhe, Germany), yeast cell suspensions were first incubated with BMTS for 10 min and washed; then, DTT or ACC was added at the same concentration and incubated for another 10 min. All permeability measurements were done in a stopped-flow apparatus (SFM-300; BioLogic, Claix, France) at 20 °C using a total injection volume of 150 µL at a flow rate of 16 mL s^−1^. The cell suspensions were mixed with an equal volume of the same buffer that additionally contained 600 mM sucrose or 60 mM glycerol, respectively, generating 300 mM or 30 mM transmembrane gradients. Changes in the intensity of 90° light scattering were monitored at 524 nm indicating cell shrinkage by osmotic water efflux and cell re-swelling by glycerol influx.

Determination of rate constants. The light scattering traces from nine technical replicates from each of two to three biological replicates (indicated in the figure legends) were averaged and normalized for each experimental condition. Rate constants of water permeability were determined in hypertonic saccharose buffer by single exponential fitting, or from combined water and glycerol permeability assays using hypertonic glycerol buffer (Sigma Plot, version 14.5, Systat Software, Frankfurt, Germany). For the latter, two rate constants were determined from double exponential fittings with one derived from the initial rapid phase representing water permeability, and the second derived from the following slower phase representing glycerol influx.

Statistical evaluation. Bar graph data are represented as mean ± S.E.M. Statistical significance of water and glycerol permeability by the cysteine modifiers was calculated by unpaired, one-tailed Student’s *t*-tests. Levels of *p* < 0.05 were assumed significant and labeled by an asterisk in the graphs, and borderline significance with *p* ≈ 0.05 is indicated by a hash symbol; non-significance by n.s.

## 3. Results

### 3.1. Methanethiosulfonate Reagents Inhibit AQP1 via Cys189 Modification

Initially, we tested whether Cys-modifying MTS reagents in principle constitute an alternative to the established mercurial pCMBS for inhibiting the water permeability of AQP1 (Figure 2A,C). Therefore, we expressed AQP1 in *Saccharomyces cerevisiae* yeast and assayed water channel functionality by stopped-flow light scattering in an osmotic outward gradient generated by 300 mM saccharose. Here, cell shrinkage due to water efflux increases the intensity of the light-scattering signal. AQP1-expressing cells (Figure 2A, blue trace) exhibited more rapid efflux kinetics than non-expressing cells (Figure 2A, black trace). The addition of 30 µM pCBMS to the AQP1-expressing cells 10 min prior to the assay (Figure 2A, magenta trace) decreased water permeability to the background level (Figure 2C).

The MTS reagents were first tested at 1 mM and 10 min pre-incubation time with AQP1-expressing yeast (Figure 2B). All led to an inhibition of AQP1 water permeability, yet to different levels (Figure 2C). The inhibitory effect depended on the size of the transferred moiety (Figure 2D). While the transfer of a small S-methyl from MMTS had a marginal effect on the water permeability, modification with S-*n*-propyl from PMTS conferred intermediate inhibition, and the S-benzyl group of BMTS that is similar in size to pCMBS fully inhibited AQP1 water permeability (Figure 2C,D). Hence, BMTS may be used as an alternative to the mercurial if full blockage of AQP1 is requested. Cys-modification by smaller MTS reagents additionally allows for stepwise modulation of AQP1 functionality.

To make sure that Cys189 of AQP1 is targeted by the modifiers, we used an AQP1 Cys189Ala variant. This mutant is fully functional when expressed in yeast (Figure 3, filled blue bar). Preincubation of AQP1 Cys189Ala-expressing cells with the MTS reagents (1 mM) or pCMBS (30 µM) for 10 min maintained full water permeability (Figure 3, open bars), confirming Cys189 of AQP1 as the interaction site.

Next, we evaluated the reaction conditions for modification of AQP1 Cys189 by BMTS for optimization of the modification reaction conditions. We varied the concentration of BMTS at 10 min incubation time (Figure 4A). Here, a concentration of 30 µM BMTS already caused some observable inhibition of AQP1, and full inhibition was achieved at 700 µM in the used 10 min time frame. From the obtained data, we calculated an IC_50_ of 111 ± 3 µM. As irreversible inhibitors, the MMTS reagents interact with its target cysteine in a time-dependent fashion towards completion rather than a concentration-dependent equilibrium. Accordingly, the concentration required for full inhibition should decrease with elongated incubation times. To test for this, we used the 400 µM BMTS concentration, which resulted in an intermediate inhibition of AQP1 water permeability after 10 min pre-incubation, and prolonged the reaction time (Figure 4B). Indeed, completion of the Cys-modifying reaction occurred after 30 min.

Based on this evaluation, we decided to use 700 µM of BMTS and 10 min incubation as suitable conditions for the following experiments.

### 3.2. Acetylcysteine and Dithiothreitol Undo the AQP1 Cys189-Modification by BMTS

A disulfide bond is formed between the S-alkyl moiety of the MTS reagent (Figure 2D) and the thiol of the modified cysteine. The addition of thiol-containing molecules such as ACC or DTT can break disulfide bonds and, thus, should be able to reverse the AQP1 Cys189-modification. To test for reversibility, we treated AQP1-expressing yeast (Figure 4C, filled blue bar) first with 700 µM BMTS for 10 min (Figure 4C, open bar), washed the cells, and added 700 µM ACC or DCC (Figure 4C, striped bars) for another 10 min. Under these conditions, both thiol reagents reversed the AQP1 Cys189-modification and re-established full water permeability.

### 3.3. BMTS Blocks Water and Glycerol Permeability of Native Erythrocytes

In order to probe the suitability of MTS reagents for their use in native mammalian cells, we chose mouse erythrocytes. Erythrocytes are characterized by high water permeability due to a high copy number of AQP1 water channels in the plasma membrane [32]. In addition to AQP1, an aquaglyceroporin is present conferring glycerol permeability, i.e., AQP3 in humans [33] and AQP9 in mice [34]. To assay for water and glycerol permeability in the same experimental setup, we exposed mouse erythrocytes to a hypertonic inward glycerol gradient of 30 mM (Figure 5A, black trace). The application of a smaller gradient than in the yeast assays was necessary because erythrocytes are much more delicate than the cell wall-protected yeast cells. Under these conditions, the erythrocytes showed the typical behavior of an initial rapid water efflux and slower re-swelling due to the osmotic effect induced by a glycerol influx. Respective rate constants for the rapid water efflux phases and the following glycerol influx phase were calculated from double-exponential fittings and are shown in Figure 5B,C. Incubation with 700 µM BMTS for 10 min inhibited both water and glycerol permeability (Figure 5A,C, orange).

### 3.4. Inhibition of Yeast-Expressed AQP9 Water and Glycerol Permeability by BMTS

Since the AQP9 channel carries three cysteine residues close to the extracellular entry site (Figure 1), we went back to the heterologous yeast expression system for further analysis. As expected, AQP9 water and glycerol permeability were strongly inhibited by BMTS and pCMBS when assayed in a hypertonic inward glycerol gradient of 30 mM (Figure 6A,B).

The location of Cys213 in the AQP9 channel corresponds to Cys189 in AQP1 (Figure 1). Accordingly, Cys213 was proposed to act as the mercurial binding site [12]. Yet, experimental data on this assumption are missing. Therefore, we generated an AQP9 Cys213Ala variant for expression in yeast. The mutant exhibited normal water and glycerol permeability (Figure 7, filled red bars).

Contrary to AQP1 Cys189Ala, though, treatment of AQP9 Cys213Ala with BMTS or pCMBS failed to generate full resistance to the Cys-modifying agents (Figure 7, open bars). The water permeability of AQP9 Cys213Ala was still inhibited to the same degree as that of the wildtype protein (compare Figure 6A and Figure 7A). The glycerol permeability of AQP9 Cys213Ala was less affected by the modifiers compared to wildtype but still remained below that of the untreated channel with only borderline significance for pCMBS treatment (compare Figure 6B and Figure 7B). This indicates that other cysteine residues in addition to Cys213, possibly Cys41 or Cys43 (Figure 1), are accessible for Cys-modifying reagents.

Together, our data show that Cys-modifying MTS reagents constitute a class of alternative inhibitors for mercurial-sensitive AQPs. They enable size-dependent gradual and reversible blockage.

## 4. Discussion

Mercury is recognized as a significant environmental risk due to its high toxicity, persistence, and bioaccumulation [27]. In aquatic ecosystems, mercury contamination can harm fish and amphibians, among others. In addition to ingestion, uptake into the human organism occurs by inhalation or via skin exposure with organomercurials being particularly prone to pass the protective skin barrier. Systemic mercury interferes with transmembrane transport and several enzymatic systems, and primarily affects the nervous system, kidneys, and reproductive system, leading to a range of severe health issues, including neurological disorders and impaired cognitive function [35]. Hence, it is appropriate to turn to non-toxic alternatives to the use of mercury when studying protein structure–function relationships by covalent modification of cysteine residues. MTS reagents are considered relatively harmless chemicals. Due to their high reactivity, which is required to fulfill their purpose as thiol-modifying agents, the compounds have a short half-life. Considering the precautions and low common quantities of reagents in a research lab, the risk to human health and the environment is moderate.

MTS reagents are well established in basic protein structure–function evaluation since the 1970s [36]. Their reactivity has been characterized to be largely independent of the structure of the transferred group, to occur rapidly under mild conditions in aqueous buffers that are non-destructive to proteins, to be highly selective for cysteine sidechains, to complete conversion at little excess of reagent (if sufficiently long reaction times are provided), and to be fully reversible by treatment with thiols, such as DTT.

A large diversity of transferrable groups in terms of chain length and volume, polarity, presence of charged functional groups, and fluorescence or spin labels is available offering many more experimental options than the use of the heavy metal mercury for Cys-modification. The application of MTS reagents is common in biochemical enzyme characterization [36] but also in transmembrane transport [37,38,39,40,41]. One can make use of cysteine positions that are naturally present in a protein of interest as in the here investigated AQP1 and AQP9, or cysteines can be inserted at desired positions in the protein using site-directed mutagenesis of the respective encoding DNA open reading frame. The presence of multiple cysteine residues, however, may lead to the modification of several sites depending on their accessibility by the MTS reagent [37]. We also observed this here in the case of AQP9 with Cys41, Cys43, and Cys213 located in the water and glycerol channel path (Figure 1). Cysteines that are not present at external or internal protein surfaces probably will not give rise to modification by Cys modifiers. An example is the aquaglyceroporin of the malaria parasite *Plasmodium falciparum*, PfAQP, which carries six cysteines, two in intracellular loops and four in transmembrane helices [38]. Yet, PfAQP is insensitive to mercurial treatment.

The various MTS reagents may further differ in their capability to penetrate cell membranes. While lipid bilayers are practically impermeable to MTS reagents that carry large or charged transfer groups, smaller types in particular MMTS may pass the membrane considerably and may elicit effects by binding to intracellularly accessible cysteine residues [37]. One study on liposomes showed the impermeability of the lipid bilayer for 2-sulfonatoethyl methanethiosulfonate, MTSES, i.e., an MTS reagent with a strongly acidic sulfonate group [39]. Likewise, trimethylaminoethyl methanethiosulfonate, MTSET, with a permanent positive charge was unable to enter the liposomes. However, the basic ethylamine methanethiosulfonate, MTSEA (pK_a_ 10.7), reached the liposome lumen quite readily despite the fact that 99% of the molecules are positively charged at physiological pH [39].

A few examples are given below to illustrate the use of MTS reagents in transmembrane transporter and channel research. We used MMTS, PMTS, and BMTS before with the human monocarboxylate transporter 1, MCT1 [37]. By introducing S-alkyl moieties, we identified a naturally occurring cysteine residue, Cys159, in the center of a hinge region enabling the rigid-body rotation during the alternating access cycle of the transporter. The MTS reagents acted in a wedge-like fashion inhibiting transport functionality depending on the size of the introduced modifier. Similar to the here observed action on AQP1, the small MMTS blocked lactate transport via MCT1 only marginally, whereas BMTS led to full inhibition; PMTS elicited an intermediate effect. It is thinkable that the identified Cys159 of MCT1 may be used for covalent modification via MTS reagents to introduce fluorescent probes as sensors for changes in the electrostatic environment or proton availability during binding of the lactate substrate and proton co-substrate for insights into the transport mechanism.

An early study on the topology of the serotonin transporter identified three endogenous cysteine positions at the extracellular side of the cell membrane: Cys109, Cys200, and Cys209 [40]. The authors exchanged the cysteines individually and in different combinations, and applied Cys-modifying MTS reagents before measuring transport activity. One finding was that replacement of Cys200 by serine increased the MTS reactivity possibly by allowing access to Cys209 providing evidence for the presence of a stabilizing disulfide bond between Cys200 and Cys209 in the wildtype transporter.

The different membrane permeability properties of the MTSES/MTSET/MTSEA reagents, as laid out above [39], may appear as a complication. On the other hand, it allowed for the determination of the topology of the Shaker potassium channel [39]. Using the wildtype channel and a variant with an exchange at the intracellular Cys391 site in combination with membrane-permeable and impermeable MTS, reagents modulated ion currents in a way that allowed for a conclusion on the topological membrane insertion of the Shaker protein.

The acidic sulfonate group carrying MTSES reagent was also used in the AQP field [41]. In the study, cysteine residues were inserted at specific sites into an engineered Cys-less variant AQP1 in which the natural cysteines had been replaced by alanine. The introduction of negative charges via the MTSES Cys modifier was found to modulate cGMP-dependent cation currents that were observed previously in AQP1 under certain experimental conditions.

## 5. Conclusions

In this study, we show that MTS reagents act as inhibitors of mercurial-sensitive AQPs. The reversibility by adding ACC or DTT provides an opportunity to restore the wildtype functionality of the investigated protein. This way, different assay conditions can be tested with the same cell preparation, eliminating deviations derived from varying expression levels between experiments. Smaller-sized modifiers, such as PMTS, can be used to achieve intermediate AQP inhibition. BMTS fully blocks AQP permeability and is suitable to replace the highly toxic mercury for inhibiting mercurial-sensitive AQPs. The large variety of available MTS reagents further allows for the post-translational introduction of diverse chemical functionalities or sensor properties into AQPs for studies of structure–function relationships.

## Figures and Tables

**Figure 1 cells-12-01742-f001:**
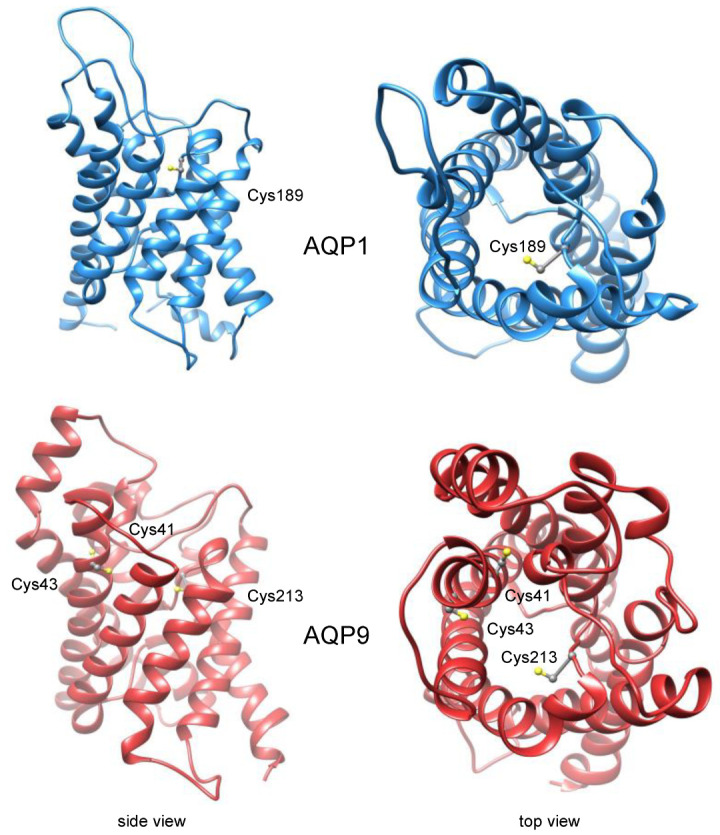
Position of cysteine residues in the channel path of mammalian water-selective AQP1 (blue; PDB# 1J4N [13]) and mammalian aquaglyceroporin AQP9 (red; model based on PDB# 1FX8 [6]).

**Figure 2 cells-12-01742-f002:**
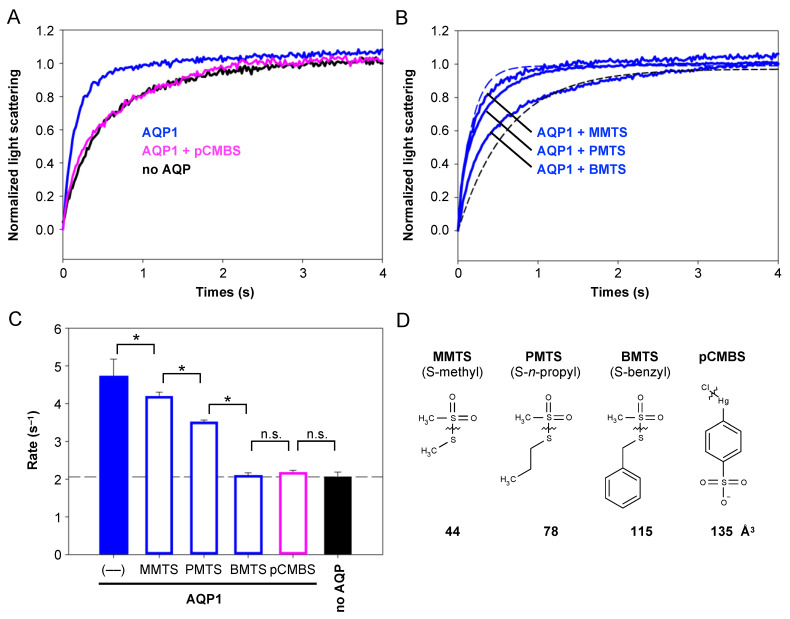
Inhibition of AQP1 water permeability by MTS reagents. (**A**) Shown is water efflux in a 300 mM saccharose outward osmotic gradient of AQP1-expressing walled yeast in the absence (blue) and presence of 30 µM pCMBS (magenta), and non-expressing yeast (black). (**B**) MTS reagents at 1 mM inhibit AQP1 water permeability depending on the molecular size of the Cys modifier. The positions of the light scattering traces from uninhibited AQP1 and non-expressing yeast are indicated by blue and black dashed lines, respectively. (**C**) Rate constants of the traces shown in (**A**) and (**B**). The error bars denote S.E.M. from three biological replicates with nine technical repetitions each. Significance is indicated by an asterisk (*p* < 0.05), and non-significance by n.s. (**D**) Molecular structures of MTS reagents and pCMBS. The transferred moiety is indicated by the zigzag line, and the van der Waals volume is shown below.

**Figure 3 cells-12-01742-f003:**
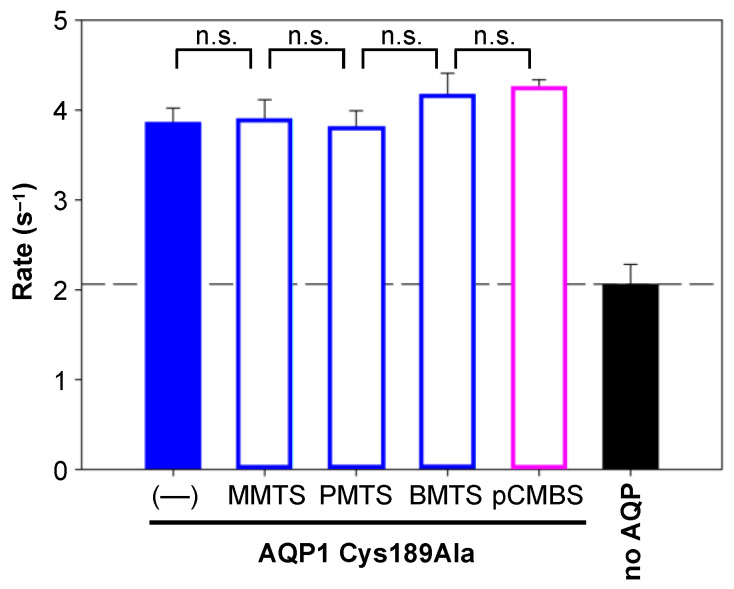
An AQP1 Cys189Ala variant is resistant to modification by MTS reagents (1 mM) and pCMBS (30 µM). The error bars denote S.E.M. from three biological replicates with nine technical repetitions each. Non-significance is indicated by n.s.

**Figure 4 cells-12-01742-f004:**
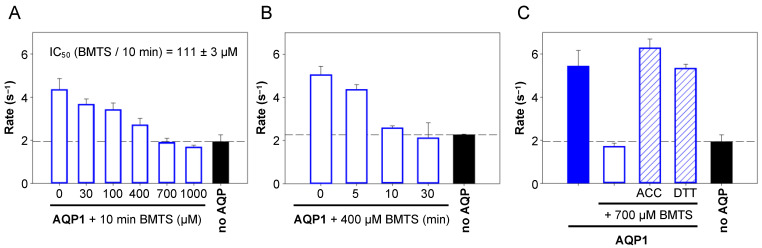
Evaluation of the reaction conditions for AQP1 Cys189 blockade by BMTS (**A**) concentration, (**B**) incubation time (400 µM BMTS), and reversibility of the modification by the thiol reagents ACC and DTT (700 µM) (**C**). The error bars denote S.E.M. from two biological replicates with nine technical repetitions each.

**Figure 5 cells-12-01742-f005:**
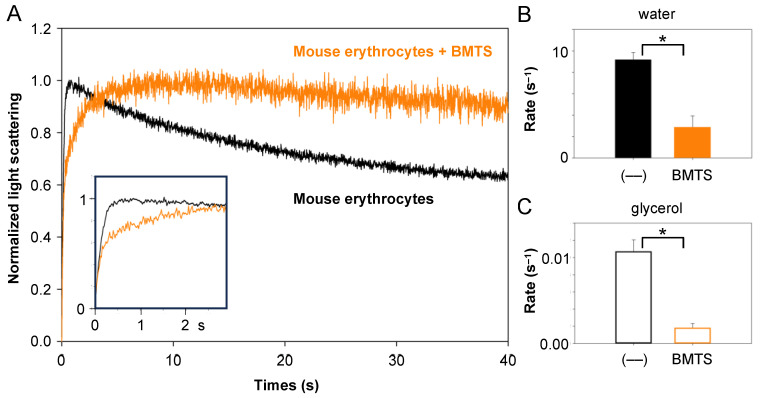
Inhibition of mouse erythrocyte water and glycerol permeability by BMTS. (**A**) Exposure of mouse erythrocytes to a hypertonic 30 mM inward glycerol gradient elicited rapid osmotic water efflux (see inset for first 2 s interval) and followed by slower glycerol influx (black trace). Treatment of the cells with BMTS inhibited water and glycerol permeability (orange trace). (**B**,**C**) Shown are the rate constants for water and glycerol permeability of untreated and BMTS-treated erythrocytes (700 µM). The error bars denote S.E.M. from three biological replicates with nine technical repetitions each. Significance is indicated by an asterisk (*p* < 0.05).

**Figure 6 cells-12-01742-f006:**
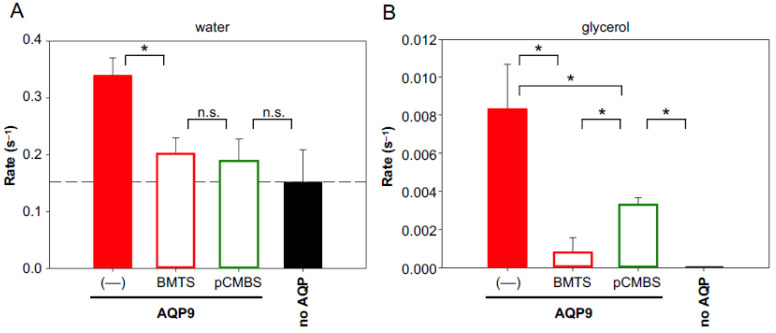
Inhibition of yeast-expressed AQP9 water (**A**) and glycerol permeability (**B**) by 700 µM BMTS and 30 µM pCMBS in a hypertonic 30 mM inward glycerol gradient. The error bars denote S.E.M. from three biological replicates with nine technical repetitions each. Significance is indicated by an asterisk (*p* < 0.05), and non-significance by n.s.

**Figure 7 cells-12-01742-f007:**
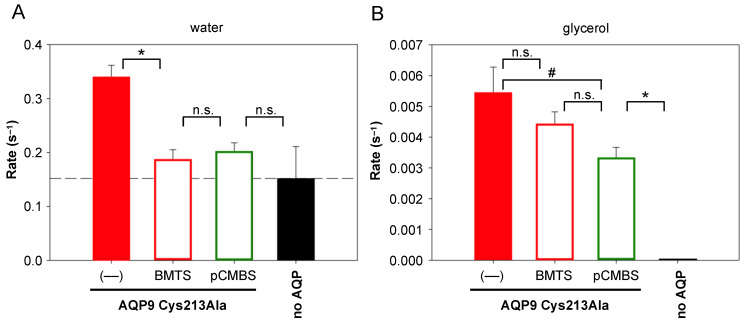
Inhibition of water (**A**) and glycerol permeability (**B**) of a yeast-expressed AQP9 Cys213Ala variant by 700 µM BMTS and 30 µM pCMBS in a hypertonic 30 mM inward glycerol gradient. The error bars denote S.E.M. from three biological replicates with nine technical repetitions each. Significance is indicated by an asterisk (*p* < 0.05), borderline significance by a hash symbol (*p* ≈ 0.05), and non-significance by n.s.

## Data Availability

All data of this study are available within the paper.

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
