# Peer review of "Methylthiosulfonate-Based Cysteine Modifiers as Alternative Inhibitors of Mercurial-Sensitive Aquaporins"

_cells, 2023, doi:10.3390/cells12131742_

Round 1

Reviewer 1 Report

Jeuken et al present a very interesting and well-designed experimental study, investigating the inhibitory effect of methylthiosulfonate reagents on AQP1 and AQP9 permeability to water and glycerol. The results are very clear, and the conclusions are well supported.

A few points to improve the study:

1 - Preparation of yeast cells for permeability experiments: protoplasts or walled cells? This should be referred to in methods.

2 - Rate constants were determined by single or double-exponential fitting. In the case of double exponential, which rate constant (1st or 2nd) was used? How can these be combined and compared with a single exponential in the same bar graphs?

3 - Fig 1B – include the traces for AQP1 and no AQP to evidence the intermediate inhibition.

4 - The IC50 for BMTS could be calculated (Fig 4A) to allow comparing the inhibitor’s potency with others reported in the literature.

5 - If the authors used 700 μM BMTS for 10 min for reversibility experiments (demonstrated in Fig.4A), which is the added value of Fig4B with BMTS 400 μM?

6 - Statistical analysis is missing in all bar graphs; in particular, information in Fig 7 would be improved.

7 - The discussion could be expanded to include considerations on the usefulness of BMTS or other MTT reagents for aquaporin research, regarding selectivity and potential in vivo applications.

Author Response

Reviewer 1:

Jeuken et al present a very interesting and well-designed experimental study, investigating the inhibitory effect of methylthiosulfonate reagents on AQP1 and AQP9 permeability to water and glycerol. The results are very clear, and the conclusions are well supported.

A few points to improve the study:

1 - Preparation of yeast cells for permeability experiments: protoplasts or walled cells? This should be referred to in methods.

>>> Walled cells without protoplastation were used. This is now clearly stated in the Methods and in the Results text.

2 - Rate constants were determined by single or double-exponential fitting. In the case of double exponential, which rate constant (1st or 2nd) was used? How can these be combined and compared with a single exponential in the same bar graphs?

>>> Single-exponential fittings for water permeability were done with light scattering traces generated in hypertonic saccharose, double-exponential fittings were done for combined water and glycerol permeability assays in hypertonic glycerol buffer. Here, the rate constant of the initial rapid phase corresponds to water permeability, and the one from the following slower phase for glycerol permeability. The calculation of rate constants is now explained in a separate subsection in the Methods.

3 - Fig 1B – include the traces for AQP1 and no AQP to evidence the intermediate inhibition.

>>> Done – we now show the positions of the traces from Fig. 1A as dashed lines in Fig. 1B for better comparison.

4 - The IC50 for BMTS could be calculated (Fig 4A) to allow comparing the inhibitor’s potency with others reported in the literature.

>>> Done – one has to keep in mind, though, that the MTS reagents are covalent binders. That means that with prolonged incubation times the IC50 value will change to lower concentrations as the compounds react towards completion rather than an equilibrium. This is now explained in the text.

5 - If the authors used 700 μM BMTS for 10 min for reversibility experiments (demonstrated in Fig.4A), which is the added value of Fig4B with BMTS 400 μM?

>>> The 400 µM condition was used to show the effect of extended incubation with a covalent modifier, see point 4 above. This is now more explicitly detailed in the text.

6 - Statistical analysis is missing in all bar graphs; in particular, information in Fig 7 would be improved.

>>> Statistics was included in the figures were relevant and a new subsection was added to the Methods on the statistical evaluation.

7 - The discussion could be expanded to include considerations on the usefulness of BMTS or other MTT reagents for aquaporin research, regarding selectivity and potential in vivo applications.

>>> The Discussion was greatly extended providing more information on the reactivity, selectivity, and use – shown as expamples from the transmembrane transport field – of the MTS reagents.

Reviewer 2 Report

The authors identify new blockers of AQPs. Overall, the manuscript is interesting and the data are well presented.

However, for all experiments it must be indicated the number of repetitions (n number) and statistical analyses have to be carried out. In all figures the concentration of the compounds needs to be indicated.

The authors discuss the replacement of mercury with their new compounds. This discussion is unclear. Do the authors want to replace mercury all over the world or just inhibitors of AQPs for experimental approaches. This discussion should be extended. The potential usage of MTS compounds in experiments should also been extended by discussion potential side effects. The concentration the authors used appear quite high. What is known about effects of MTS compounds on cells if used in such concentrations? How about specificity?

Author Response

Reviewer 2:

The authors identify new blockers of AQPs. Overall, the manuscript is interesting and the data are well presented.

However, for all experiments it must be indicated the number of repetitions (n number) and statistical analyses have to be carried out. In all figures the concentration of the compounds needs to be indicated.

>>> Done – the number of repetitions, compound concentrations, and statistics were added to all figures.

The authors discuss the replacement of mercury with their new compounds. This discussion is unclear. Do the authors want to replace mercury all over the world or just inhibitors of AQPs for experimental approaches.

>>> We are sorry for the unclear wording, which was corrected to specifically refer to inhibition of mercurial-sensitive AQPs.

This discussion should be extended. The potential usage of MTS compounds in experiments should also been extended by discussion potential side effects. The concentration the authors used appear quite high. What is known about effects of MTS compounds on cells if used in such concentrations? How about specificity?

>>> The Discussion was greatly extended providing more information on the reactivity, selectivity, and use – shown as expamples from the transmembrane transport field – of the MTS reagents, see point 7 by reviewer 1 above.

Round 2

Reviewer 1 Report

The authors replied to all comments and improved the manuscript.

Figure 6A (in the ms revised version) lost the bars and needs to be fixed.

Author Response

Figure 6A (in the ms revised version) lost the bars and needs to be fixed.

>>> The figure (6A) is displayed correctly when I view the files. Also the journal QC and second reviewer did not notice the issue. Maybe this is related to Reviewer 1's computer? If there is a problem with Fig. 6A I ask the journal editors to get back to me to resolve the point.